# A Photoluminescence Study of Eu^3+^, Tb^3+^, Ce^3+^ Emission in Doped Crystals of Strontium-Barium Fluoride Borate Solid Solution Ba_4−*x*_Sr_3+*x*_(BO_3_)_4−*y*_F_2+3*y*_ (BSBF)

**DOI:** 10.3390/ma16155344

**Published:** 2023-07-29

**Authors:** Tatyana B. Bekker, Alexey A. Ryadun, Sergey V. Rashchenko, Alexey V. Davydov, Elena B. Baykalova, Vladimir P. Solntsev

**Affiliations:** 1Sobolev Institute of Geology and Mineralogy, Siberian Branch of the Russian Academy of Sciences, 630090 Novosibirsk, Russia; rashchenkos@gmail.com (S.V.R.); davydov.av@gmail.com (A.V.D.); solntsev@igm.nsc.ru (V.P.S.); 2Department of Geology and Geophysics, Novosibirsk State University, 630090 Novosibirsk, Russia; e.baikalova@nsu.ru; 3Nikolaev Institute of Inorganic Chemistry, Siberian Branch of the Russian Academy of Sciences, 630090 Novosibirsk, Russia; ryadunalexey@mail.ru

**Keywords:** borates, dopants, rare-earth elements, photoluminescence

## Abstract

The present study is aimed at unveiling the luminescence potential of Ba_4−*x*_Sr_3+*x*_(BO_3_)_4−*y*_F_2+3*y*_ (BSBF) crystals doped with Eu^3+^, Tb^3+^, and Ce^3+^. Owing to the incongruent melting character of the phase, the NaF compound was used as a solvent for BSBF crystal growth. The structure of BSBF: Eu^3+^ with Eu_2_O_3_ concentration of about 0.7(3) wt% was solved in the non-centrosymmetric point group *P*6_3_*mc*. The presence of Eu_2_O_3_ in BSBF: Eu^3+^ leads to a shift of the absorption edge from 225 nm to 320 nm. The photoluminescence properties of the BSBF: Ce^3+^, BSBF: Tb^3+^, BSBF: Eu^3+^, and BSBF: Eu^3+^, Tb^3+^, Ce^3+^ crystals have been studied. The unusual feature of europium emission in BSBF is the intensively manifested ^5^D_0_→^7^F_0_ transition at about 574 nm, which is the strongest for BSBF: Eu^3+^ at 370 nm excitation and for BSBF: Eu^3+^, Tb^3+^, Ce^3+^ at 300 nm and 370 nm excitations. No evidence of Tb^3+^→Eu^3+^ energy transfer was found for BSBF: Eu^3+^, Tb^3+^, Ce^3+^. The PL spectra of BSBF: Eu^3+^ at 77 and 300 K are similar with CIE chromaticity coordinates of (0.617; 0.378) at 300 nm excitation and (0.634; 0.359) at 395 nm excitation and low correlated color temperature which implies application prospects in the field of lightning. Due to the high intensity of ^5^D_0_→^7^F_0_ Eu^3+^ transition at 370 nm excitation, the BSBF: Eu^3+^ emission is yellow-shifted.

## 1. Introduction

The increasing exploration of borate compounds is motivated mainly by their chemical and physical stability, a wide range of transparency from ultraviolet (UV) to infrared range, and a high laser-induced damage threshold. A number of borates demonstrate high birefringence, e.g., α-BaB_2_O_4_, Ba_2_Na_3_(B_3_O_6_)_2_F, and outstanding nonlinear optical properties for laser frequency conversion from infrared to UV- and visible ranges, e.g., β-BaB_2_O_4_, LiB_3_O_5_ [1]. Aside from optical properties, another reason for the continued attention to borates is their remarkable structural variety owing to the dual hybridization of boron atoms and the possibility for polymerization via bridging oxygen atoms.

During the past decades, the luminescence of borates doped with rare earth elements has been intensively analyzed [2,3,4,5,6]. The main areas of their possible exemplification are the components of white light-emitting diodes (WLEDs) required for modern lighting and displays and thermoluminescence phosphors for radiation dosimetry [7,8]. Among the reported borate materials for WLEDs are LiBaBO_3_: Eu^2+^, Tb^3+^, and Eu^3+^, red phosphor with tunable-color emission implying the Eu^2+^→Tb^3+^→Eu^3+^ energy transfer [9], LaSc_3_(BO_3_)_4_:Eu^3+^ red phosphor with zero thermal quenching and internal quantum efficiency of 88.3% [4], GdBO_3_: Ce^3+^, Tb^3+^, Eu^3+^ broadband-excited red phosphor [10], LiBa_12_(BO_3_)_4_F_4_: Eu^3+^, Tb^3+^, Ce^3+^ single-matrix white phosphor [11].

The main object of this study is highly unconventional non-centrosymmetric (*P*6_3_*mc*) solid solution Ba_4−*x*_Sr_3+*x*_(BO_3_)_4−*y*_F_2+3*y*_ (BSBF) exhibiting both cationic Ba^2+^↔Sr^2+^ and anionic [(BO_3_)F]^4−^↔[F_4_]^4−^ isomorphism [12]. Such structural flexibility allows tuning the optical properties, for instance, the value of the absorption edge [13]. Particular emphasis should be laid on the fact that crystals of Ba_4−_*_x_*Sr_3+*x*_(BO_3_)_4−_*_y_*F_2+3*y*_ are characterized by a property completely new to the class of borates. This is a reversible color change upon X-ray irradiation with the possibility of returning the crystals to their original uncolored state by irradiating with intense light in the range of 300–400 nm. X-ray irradiation is accompanied by the formation of induced color centers, which were studied by optical and electron paramagnetic resonance spectroscopy. The combination of properties of the BSBF crystals allows them to be used as a dose indicator [12,13]. 

The luminescent properties of undoped Ba_4−_*_x_*Sr_3+*x*_(BO_3_)_4−_*_y_*F_2+3*y*_ crystals were discussed in detail in Ref. [12]. The potential employment of fluoroborates can be enhanced by the incorporation of rare earth elements, endowing the crystals with additional luminescence properties. The present study is aimed at unveiling the luminescence potential of Ba_4−_*_x_*Sr_3+*x*_(BO_3_)_4−_*_y_*F_2+3*y*_ doped with Eu^3+^, Tb^3+^, and Ce^3+^ for application in WLEDs as single-matrix or composite phosphors. 

## 2. Materials and Experimental Methods

Crystal Growth. We grew crystals using NaF as a solvent owing to the incongruent melting of Ba_4−*x*_Sr_3+*x*_(BO_3_)_4−*y*_F_2+3*y*_; BaCO_3_, SrCO_3_, SrF_2_, H_3_BO_3_, NaF, Ce_2_(CO_3_)_3_·5H_2_O, Tb_4_O_7_, E_2_O_3_ were used as starting reagents. The growth process was carried out in a platinum crucible 40 mm in diameter in air; the weight of the initial high-temperature solution was 40 g. As it follows from the data in Table 1, the effect of the rare earth elements addition on the liquidus temperature was minimal. Crystals were grown on a platinum loop without rotation and pulling at a cooling rate of 2 °C per day for approximately 15 days. The average weight of the crystals was about 8 g. We included the photographs of 1 mm thick plates made of the grown crystals in Table 1.

Structure solution. To assess the influence of rare earth elements on the structure, we completed a single crystal X-ray diffraction analysis of BSBF: Eu^3+^, grown from a high-temperature solution with a relatively high europium concentration of 3.5 wt%, using a STOE IPDS diffractometer with graphite-monochromatized MoKα radiation and image plate detector. The ESPERANTO protocol, CrysAlisPro [14], SUPERFLIP, and Jana2020 software [15,16] were used for row data analysis and subsequent refinement of the structure.

Optical Spectroscopy. Transmission spectra of 1 mm thick plates were recorded with UV-VIS-NIR spectrometer sUV-3101 PC, Shimadzu.

The study of photoluminescence properties and lifetime measurements was performed with a Fluorolog 3 (Horiba Jobin Yvon, Kyoto, Japan) spectrofluorometer. Optistat DN was used to investigate the luminescence property at low temperatures. The measurement of the quantum yield was performed using a G8 (GMP SA, Zurich, Switzerland) spectralon-coated integrating sphere connected to a Fluorolog 3 spectrofluorimeter. More details on the equipment used are provided in Ref. [11].

X-ray powder diffraction (XRD) analysis. X-ray powder diffraction analysis was carried out using DRON 8 (Russia) with CuKα (1.5418 Å) radiation, Mythen2 R1 D detector (Switzerland) with a step width of 0.1° and 5 s of exposure time per position.

Energy-dispersive X-ray (EDX) microanalysis. In order to estimate europium and terbium concentration in grown crystal, the samples were examined on an MIRA 3 LMU scanning electron microscope (Tescan Orsay Holding, Brno, Czech Republic) in combination with an INCA 450 energy dispersive X-ray microanalysis system. It is worth noting that due to the intersection of the Lα lines for barium (4.84 keV) and cerium (4.47 keV), measurements of the cerium concentration were not possible. The detection (three sigma) limit for rare earth element measurements was 0.51 wt%.

## 3. Results and Discussion

Crystal Structure. The results of X-ray single crystal analysis show that BSBF: Eu^3+^ is characterized by the same point symmetry as the undoped BSBF crystal, *P*6_3_*mc*. The details about the X-ray diffraction structural analysis and its results are presented in Appendix A and the CIF file (CCDC deposition number 2254976). The X-ray powder diffraction pattern of BSBF: Eu^3+^ is shown in Appendix A. The suggested geterovalent isomorphic scheme is as follows: 3(Ba, Sr)^2+^←2Eu^3+^ + □, □–vacancy in cationic sites. We believe that this substitution scheme is also valid for terbium and cerium ions. Refined stoichiometry of the compound ‘B_3.703662_Ba_3.172464_F_2.889012_O_11.11099_Sr_3.827532_’ may be represented as [Ba_3_Sr_3_(BO_3_)_3_](Ba_1−*x*_Sr*_x_*)[(BO_3_)_1−*y*_F_1_*_+3y_*] with *x* ≈ 0.8, *y* ≈ 0.3.

In the structure of BSBF crystal, there are three nonequivalent crystallographic cationic positions: Ba^2+^, Sr^2+^ (both with *Cs* symmetry), and isomorphic position M (*C3v* symmetry), in which barium and strontium are statistically distributed. The coordination number of Ba, Sr, and M positions is 15, 13, and 16, respectively (Figure 1). The possibility of the presence of Ba^2+^, Sr^2+^, Eu^3+^, and a vacancy in the cationic positions does not make it possible to unambiguously determine the concentration of these species when refining the structure. To assess which of the positions is the most favorable for the incorporation of Eu^3+^ ions, we used the bond valence sum method (Appendix A). Corresponding calculations taking into account the positional disorder in X1O/X1F and X2O/X2F sites were performed using bond-valence parameters R = 2.076 for Eu^3+^−O and R = 1.961 Eu^3+^−F (R = 1.961) [17]. The resulting values for Sr^2+^, Ba^2+^, and mixed M^2+^ positions are 1.8, 1.0, and 1.5, respectively (see Appendix A). As the bvs for the Sr^2+^ position is closer to the Eu^3+^ valence, this position is slightly more favorable for substitution.

Optical properties.

Transmission spectra of the plates made of doped BSBF crystals are depicted in Figure 2. The optical absorption edge of BSBF: Ce^3+^, BSBF: Tb^3+^, and BSBF: Eu^3+^, Tb^3+^, Ce^3+^ approximately coincides and corresponds to 225 nm, while BSBF: Tb^3+^ demonstrates the best transparency in the UV range. This value coincides with the absorption edge of the undoped Ba_4_Sr_3_(BO_3_)_4_F_2_ crystal, which is 225 nm (5.517 eV) for 0.5 thick plate at 300 K [13]. According to EDX microanalysis, the concentration of Tb_2_O_3_ and Eu_2_O_3_ in BSBF: Tb^3+^ and BSBF: Eu^3+^, Tb^3+^, Ce^3+^ is below the limit of detection, the concentration of Eu_2_O_3_ in BSBF: Eu^3+^ is about 0.7(3) wt%. It can be seen that an increase in the Eu^3+^ concentration in BSBF: Eu^3+^ leads to a shift of the absorption edge to the long wavelength region up to 320 nm, which might be accounted for by the high intensity of charge transfer transitions [18].

BSBF: Ce^3+^. The 5d^1^→4f^1^ luminescence of Ce ions sufficiently depends on the nature of the matrix and is in the range from the ultraviolet to the red region of the visible spectrum [19,20,21,22]. In the photoluminescence (PL) spectra of BSBF: Ce^3+^, weak nicely resolved bands corresponding to the Ce^3+ 5^D_3/2_→^2^F_7/2,_ ^2^F_5/2_ transitions are detected at 400–500 nm under 360–390 nm excitation at 77 K (Figure 3a). Transitions in the same spectral range assigned to cerium ions were reported for Ba_2_Y_5_B_5_O_17_: Ce^3+^ phosphor [21]. It is also possible to distinguish an additional band with a maximum of around 460 nm, most clearly at 360–390 nm excitation. For emission at 460 nm, the maximum of the excitation band is about 360 nm (Figure 3b). A similar broadband at 400–500 nm associated with intrinsic defects is observed in the luminescence spectra of undoped crystals at 365 nm excitation [13].

BSBF: Tb^3+^. The PL spectra of Tb^3+^ ions are due to 4f^8^-4f^8^ transitions, shielded from the host crystal field by electrons of outer 5s and 5p shells, and, therefore, practically insensitive to the matrix. Luminescent spectra of BSBF: Tb^3+^ at 300 nm excitation consist of four relatively narrow peaks arising from ^5^D_4_→^7^F_6,5,4,3_ transitions [23], located in our case at about 489, 545, 586, and 622 nm, respectively (Figure 4a). As the temperature increases from 77 K to 300 K, the intensity of PL decreases monotonically. The spectrum observed at 270 nm excitation distinctively reveals the transition from both ^5^D_3_ and ^5^D_4_ energy levels. The essential feature of the spectrum is the broadening and splitting of the bands, which is most pronounced for the ^5^D_4_→^7^F_6_ (485, 489, and 497 nm), ^5^D_4_→^7^F_5_ (538 and 545 nm), and ^5^D_4_→^7^F_4_ (580, 585, and 590 nm) transitions (Figure 4b). This may be due to the presence of Tb^3+^ ions in several cationic positions.

The PLE spectra for emission at 545 nm consist of bands with a sharp edge at around 325 nm (Figure 4c). The observed shape of the spectra is characteristic of transitions between the valence and conduction bands. In the range of 360–380 nm, the weak Tb^3+^ transitions are revealed (Figure 4c).

BSBF: Eu^3+^ and BSBF: Eu^3+^,Tb^3+^,Ce^3+^. In the case of Eu^3+^, 4f^6^-4f^6^ transitions take place between the lowest excited state ^5^D_0_ and seven multiplets of ^7^F_J_ (J = 0–6) ground term. Transitions to the ^7^F_5_ and ^7^F_6_ multiplets are at 740–770 nm and 810–840 nm, respectively [24], and often lie outside of the investigated spectral range. The PL spectra of both crystals at 300, 370, and 395 nm excitation reveal broadened and split bands, which might be caused by the presence of rare earth elements in nonequivalent crystallographic sites and crystal field splitting (Figure 5, Table 2).

The prominent feature of PL spectra at 300 nm excitation is the high intensity of the ^5^D_0_→^7^F_0_ transition at about 574 nm. Such 0-0 transitions are forbidden according to the ΔJ selection rule. The explanations proposed so far for the breakdown of this rule assume crystal filed induced J-mixing [25,26,27] and mixing of charge-transfer state [28] into the 4f^6^ wavefunctions. It is worth noting that these two mechanisms are inversely related. In Ref. [26], the authors admit that charge transfer states of Eu^3+^ lie near the 4f^6^ levels and, therefore, may sufficiently affect the electronic structure of Eu^3+^. Charge transfer bands (CTB) are due to electron transfer. This process implies the formal reduction in trivalent europium to bivalent state and is accompanied by a change in the ionic radius of europium and strong lattice relaxation. Since CTB in all Eu-containing BSBF crystals has a pronounced intensity (Figure 6), we believe that charge-transfer state mixing is the key factor causing the high intensity of the above-mentioned transition.

Among the compounds with the strong intensity of the ^5^D_0_→^7^F_0_ transition are LaOBr [29], Sr_5_(PO_4_)_3_F [30], Ca_10_(PO_4_)_6_(OH)_2_ [29,31], BaFCl [26,28]. The ratio of intensities of ^5^D_0_→^7^F_0_/^5^D_0_→^7^F_1_ listed in Ref. [26] for LaOBr, Sr_5_(PO_4_)_3_F, Ca_10_(PO_4_)_6_(OH)_2_, and BaFCl compounds is 2.5, >20, ~10, and 35, respectively. For BSBF: Eu^3+^ and BSBF: Eu^3+^, Tb^3+^, Ce^3+,^ this value is about 2.9 and 3.4, respectively. The ^5^D_0_→^7^F_1_ transition, which is of a magnetic-dipole nature, is chosen for the comparison as its intensity is nearly insensitive to site symmetry. According to Binnemans and Gorller–Warland [32], the ^5^D_0_→^7^F_0_ transition manifests itself only if Eu^3+^ ions occupy the site with the symmetry of Cnv, Cn, or Cs, which agrees with the symmetry of cationic positions in BSBF.

Transitions with J > 0 depend on the site symmetry [33]. According to Ref. [34], the number of components for ^5^D_0_→^7^F_j_ (J = 0–4) transitions is 1, 3, 5, 7, and 9 for the site with *Cs* symmetry, and 1, 2, 3, 3, and 5 for the *C3v* site. It is worth noting that the major difficulty in unambiguous assignment of the peaks is their overlap due to a small crystal field splitting. The symmetry-sensitive ^5^D_0_→^7^F_1_ of magnetic-dipole nature and ^5^D_0_→^7^F_2_ of electric-dipole nature transitions are usually taken into consideration to establish a connection between site symmetry and band splitting [24,35]. In both BSBF: Eu^3+^ and BSBF: Eu^3+^, Tb^3+^, Ce^3+^, the splitting of the bands related to these transitions is observed (Figure 5). Thus, the BSBF: Eu^3+^ crystal at 300 nm excitation exhibits three well-resolved peaks at about 586, 592, and 603 nm associated with the ^5^D_0_→^7^F_1_ transition (Figure 5a). The intensity of various transitions in the spectra of BSBF: Eu^3+^ nonmonotonically changes with temperature (Figure 5a,c,e).

In addition to the discussed peaks associated with the transitions of Eu^3+^ ions, the BSBF: Eu^3+^, Tb^3+^, Ce^3+^ crystal reveals weak peaks at around 545 nm under 300 nm excitation related to the ^5^D_4_→^7^F_5_ transition of Tb^3+^ ions (Figure 5b). The emission of cerium ions expected in the region 400–450 nm is virtually absent (Figure 5d), which differs dramatically from the luminescence spectra observed in LiBa_12_(BO_3_)_4_F_4_ (LBBF): Eu^3+^, Tb^3+^, Ce^3+^ crystal grown from the high-temperature solution with exactly the same concentration of Eu^3+^, Tb^3+^, and Ce^3+^ [11]. In the latter case, strong emission of cerium ions is observed at 77 K.

At 370 nm excitation, the most intense band is attributed to ^5^D_0_→^7^F_0_ Eu^3+^ transition in both BSBF: Eu^3+^ and BSBF: Eu^3+^, Tb^3+^, Ce^3+^ crystals. Relatively strong transitions at 489 nm and 543 nm of Tb^3+^ are present in BSBF: Eu^3+^, Tb^3+^, and Ce^3+^ (Figure 5d).

The intensity ratio of the charge transfer band and transitions in Eu^3+^ ions differs significantly for BSBF: Eu^3+^ and BSBF: Eu^3+^, Tb^3+^, Ce^3+^ (Figure 6). For BSBF: Eu^3+^ for emission at 613 nm, Eu^3+^ transitions have a dominant intensity, which is accounted for by a relatively higher concentration of Eu^3+^ in BSBF: Eu^3+^.

In order to verify the realization of energy transfer from terbium to europium, reported for a number of compounds such as La_3_GaGe_5_O_16_:Tb^3+^, Eu^3+^ [36], La_0.02_Tb_0.90_Eu_0.08_PO_4_ [37], A_3_Tb_0.90_Eu_0.10_(PO_4_)_3_ (A = Sr, Ba) [38], WO_3_: Tb^3+^*_x_*Eu^3+^*_y_* [39], we measured decay times for the BSBF: Eu^3+^, BSBF: Tb^3+^, and BSBF: Eu^3+^, Tb^3+^, Ce^3+^ crystals. The decay time for BSBF: Eu^3+^ for emission at 613 nm after excitation at 395 nm is 2.2 ms (Figure 7a). Decay times for BSBF: Tb^3+^ (Figure 7b) and BSBF: Eu^3+^, Tb^3+^, Ce^3+^ (Figure 7c) for emission at 545 nm after excitation at 300 nm are typical of terbium ions and close enough in value, 2.58 ms and 2.64 ms, respectively, which cast doubt on the implementation of energy transfer.

The CIE chromaticity coordinates and correlated color temperature of BSBF: Tb^3+^, BSBF: Eu^3+^, and BSBF: Eu^3+^, Tb^3+^, and Ce^3+^ crystals are provided in Figure 8 and Table 3. The spectra of BSBF: Eu^3+^ at temperatures of 77 K and 300 K are quite similar, which results in the close values of CIE coordinates. Due to the high intensity of ^5^D_0_→^7^F_0_ Eu^3+^ transition at 370 nm excitation, the BSBF: Eu^3+^ emission is yellow shifted (points 5, 6, Figure 8) in comparison with emission at 300 nm (points 3, 4, Figure 8) and 370 nm (points 7, 8, Figure 8) excitation. The quantum yield for BSBF: Eu^3+^ at 395 nm excitation is about 10.6%. Thus, of special interest is the dependence of luminescence intensity and quantum yield on Eu^3+^ concentration, which requires further study.

Unlike the LBBF: Eu^3+^, Tb^3+^, Ce^3+^ [10] and LBBF: Eu^3+^, Tb^3+^ [40] crystals, which exhibit white emission close to daylight at 370 nm excitation, the BSBF: Eu^3+^, Tb^3+^, Ce^3+^ emission at 370 nm excitation is yellow shifted (points 11, 12, Figure 8). This might be due to the different ability of europium, terbium, and cerium to enter the BSBF structure (as we mentioned above, the emission of cerium ions in BSBF: Eu^3+^, Tb^3+^, Ce^3+^ is practically absent). The BSBF: Eu^3+^, Tb^3+^, Ce^3+^ emission under 300 nm (points 9, 10, Figure 8) and 395 nm (points 13, 14, Figure 8) excitations are in the red area as the Eu^3+^ transitions are the most prominent.

## 4. Conclusions

The Ba_4−*x*_Sr_3+*x*_(BO_3_)_4−*y*_F_2+3*y*_ (BSBF) crystals doped with Eu^3+^, Tb^3+^, and Ce^3+^ were grown using NaF as a solvent. The structure of the BSBF: Eu^3+^ crystal was solved in the same group *P*6_3_*mc* as the undoped BSBF crystal. It reveals three nonequivalent crystallographic cationic positions: ^(15)^Ba, ^(13)^Sr, and isomorphic position ^(16)^M, statistically occupied by Ba and Sr. The presence of Eu_2_O_3_ in BSBF: Eu^3+^ in a concentration of about 0.7(3) wt% leads to a shift of the absorption edge from 225 nm to 320 nm. The results of the study of luminescence properties of BSBF: Ce^3+^, BSBF: Tb^3+^, BSBF: Eu^3+^, and BSBF: Eu^3+^, Tb^3+^, Ce^3+^ show that terbium and europium can be effectively doped into the host while cerium emission is very weak. The observed broadening and splitting of the emission peaks can be associated both with splitting in accordance with the position symmetry and with the fact that impurity atoms occupy crystallographically nonequivalent positions in the structure. Decay times for BSBF: Tb^3+^ and BSBF: Eu^3+^, Tb^3+^, Ce^3+^ for emission at 545 nm after excitation at 300 nm are rather close, 2.58 ms and 2.64 ms, respectively, which cast doubt on the implementation of Tb^3+^→Eu^3+^ energy transfer. The PL spectra of BSBF: Eu^3+^ at 77 and 300 K are similar with CIE chromaticity coordinates of (0.617; 0.378) at 300 nm excitation and (0.634; 0.359) at 395 nm excitation and low correlated color temperature which implies application prospects in the field of lighting. Due to the high intensity of ^5^D_0_→^7^F_0_ Eu^3+^ transition at 370 nm excitation, the BSBF: Eu^3+^ emission is yellow-shifted. The quantum yield for BSBF: Eu^3+^ at 395 nm excitation is about 10.6%. The dependence of the luminescence intensity and quantum yield on the concentration of rare earth elements in BSBF requires further study.

## Figures and Tables

**Figure 1 materials-16-05344-f001:**
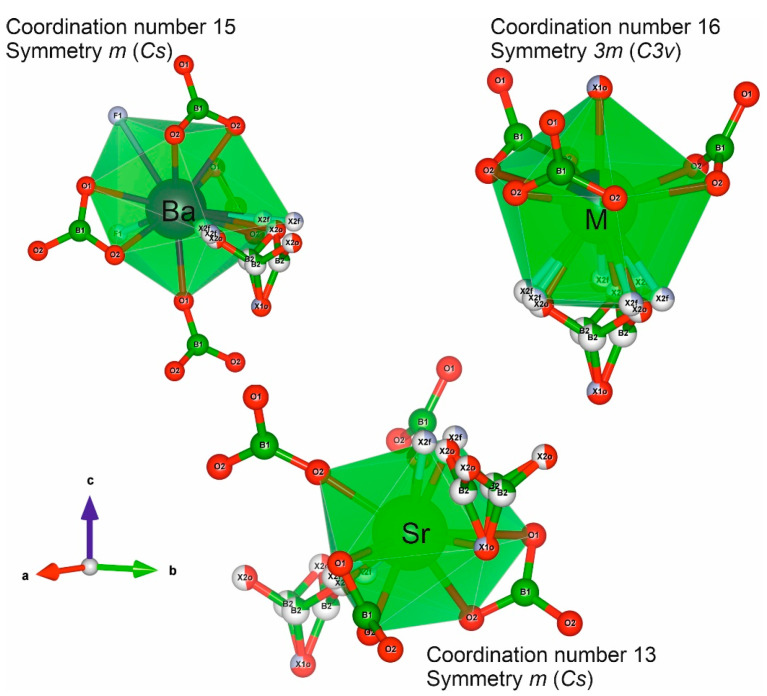
Coordinated environment of three different cationic positions, Ba, Sr, and M, in the Ba_4−*x*_Sr_3+*x*_(BO_3_)_4−*y*_F_2+3*y*_ (*P*6_3_*mc*) solid solution. Position M is statistically populated with Ba and Sr.

**Figure 2 materials-16-05344-f002:**
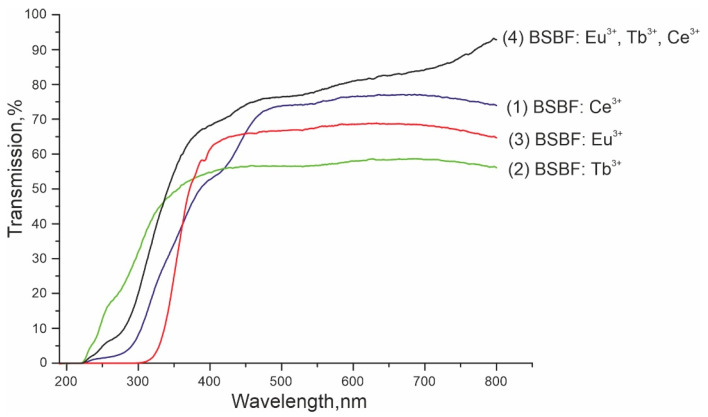
Transmission spectra of 1 mm thick plates made of doped Ba_4−*x*_Sr_3+*x*_(BO_3_)_4−*y*_F_2+3*y*_ (BSBF) crystals: (1) BSBF: Ce^3+^, (2) BSBF: Tb^3+^, (3) BSBF: Eu^3+^, (4) BSBF: Eu^3+^, Tb^3+^, Ce^3+^. Photographs of the corresponding plates are given in Table 1.

**Figure 3 materials-16-05344-f003:**
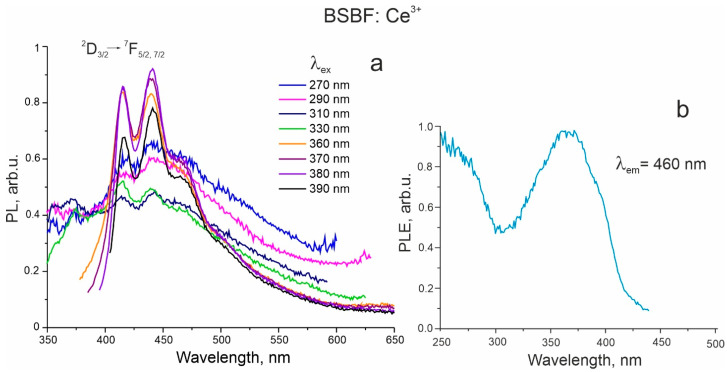
PL spectrum of BSBF: Ce^3+^ under excitation in 270–390 nm range (**a**) and PLE spectrum of BSBF: Ce^3+^ for emission at 460 nm (**b**), 77 K.

**Figure 4 materials-16-05344-f004:**
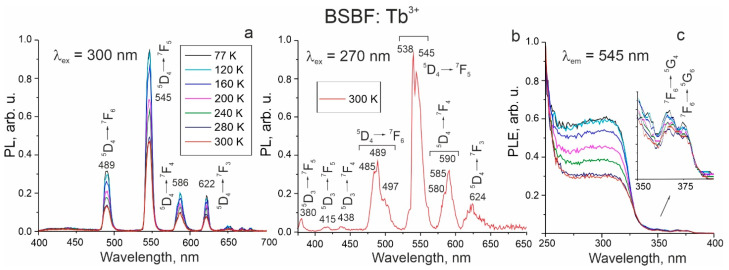
PL spectra of BSBF: Tb^3+^ crystal at 300 nm (**a**) and 270 nm (**b**) excitation and PLE spectra for emission at 545 nm (**c**). Spectra coloring for (**c**) is the same as it is in (**a**).

**Figure 5 materials-16-05344-f005:**
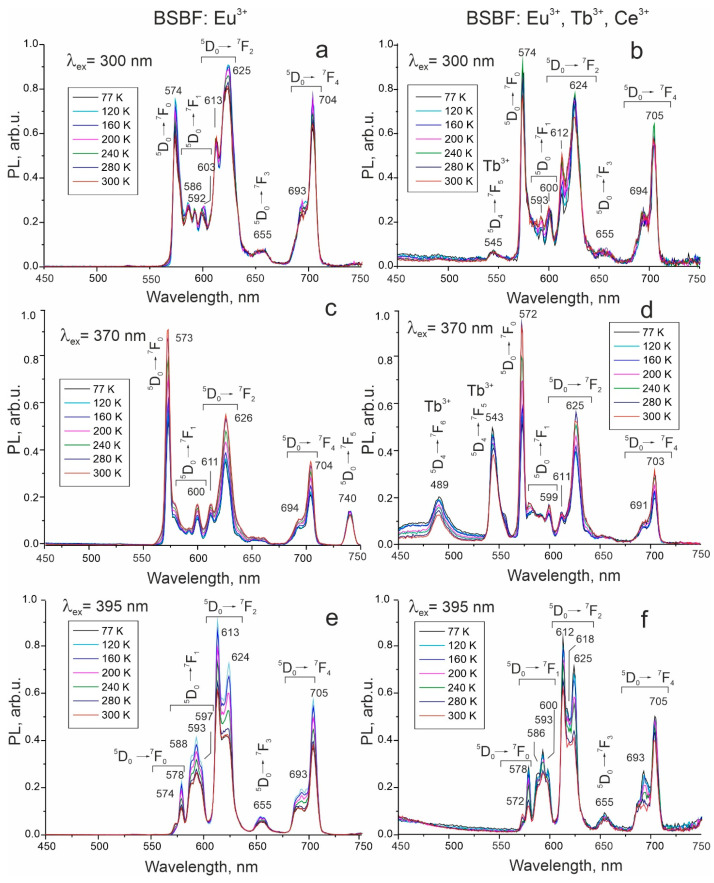
PL spectra of BSBF: Eu^3+^ at 300 nm (**a**), 370 nm (**c**), and 395 nm (**e**) excitation and PL spectra of BSBF: Eu^3+^, Tb^3+^, Ce^3+^ at 300 nm (**b**), 370 nm (**d**), and 395 nm (**f**) excitation.

**Figure 6 materials-16-05344-f006:**
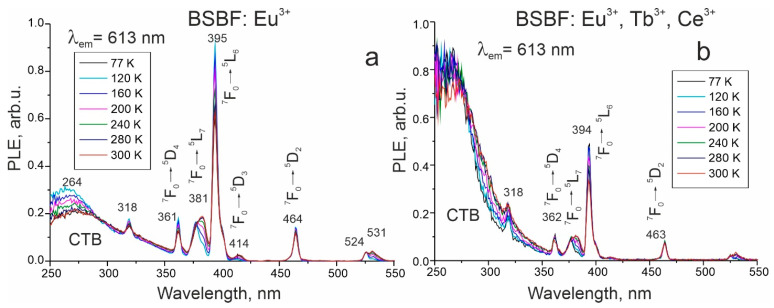
PLE spectra of BSBF: Eu^3+^ (**a**) and BSBF: Eu^3+^, Tb^3+^, Ce^3+^ (**b**) for emission 613 nm, CTB—charge transfer band.

**Figure 7 materials-16-05344-f007:**
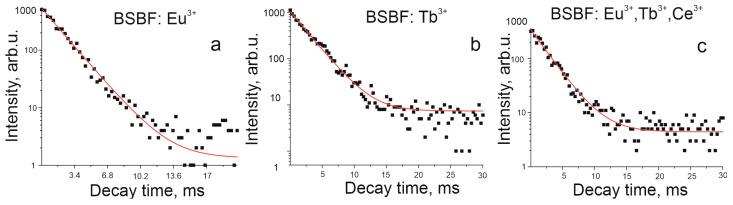
Decay curves for BSBF: Eu^3+^ at 613 nm emission under excitation at 395 nm (**a**); BSBF: Tb^3+^ at 545 nm emission under excitation at 300 nm (**b**), and BSBF: Eu^3+^, Tb^3+^, Ce^3+^ crystal at 545 nm emission under excitation at 300 nm (**c**), 300 K.

**Figure 8 materials-16-05344-f008:**
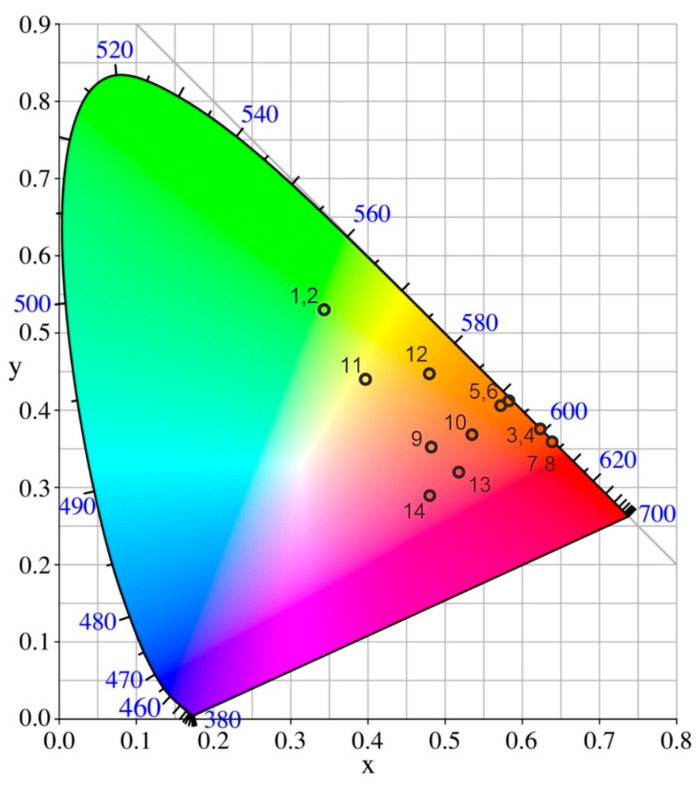
CIE chromaticity diagram for BSBF: Tb^3+^ at 300 nm excitation at (1) 77 and (2) 300 K; BSBF: Eu^3+^ at 300 nm excitation at (3) 77 K and (4) 300 K, at 370 nm excitation at (5) 77 K and (6) 300 K, and at 395 nm excitation at (7) 77 K and (8) 300 K; BSBF: Eu^3+^, Tb^3+^, Ce^3+^ at 300 nm excitation at (9) 77 K and (10) 300 K, at 370 nm excitation at (11) 77 K and 300 K (12), and at 395 nm excitation at (13) 77 K and (14) 300 K. Corresponding CIE chromaticity coordinates and correlated color temperature are given in Table 3.

**Table 1 materials-16-05344-t001:** Compositions of high-temperature solutions used for Ba_4−*x*_Sr_3+*x*_(BO_3_)_4−*y*_F_2+3*y*_ (BSBF) crystal growth.

CompositionCrystal	Ba_3_Sr_4_(BO_3_)_4_F_2_(mol%)	NaF(mol%)	Eu^3+^ (wt%)	Tb^3+^(wt%)	Ce^3+^(wt%)	T_liq_(°C)	Plate Made of Grown Crystal
BSBF: Ce^3+^	75	25	−	−	0.4	998	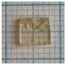
BSBF: Tb^3+^	75	25	−	0.25	−	992	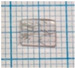
BSBF: Eu^3+^	75	25	3.5	−	−	999	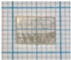
BSBF: Eu^3+^, Tb^3+^, Ce^3+^	75	25	0.1	0.25	0.4	998	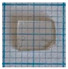

**Table 2 materials-16-05344-t002:** Observed bands in the BSBF: Eu^3+^ and BSBF: Eu^3+^, Tb^3+^, Ce^3+^ crystals under 300 nm, 370 nm, and 395 nm excitations.

	Band Maximum, nm					
	BSBF: Eu^3+^			BSBF: Eu^3+^, Tb^3+^, Ce^3+^		
	Excitation Wavelength, nm			Excitation Wavelength, nm		
	300	370	395	300	370	395
^5^D_4_→^7^F_6_ Tb^3+^	−	−	−	−	489	−
^5^D_4_→^7^F_5_ Tb^3+^	−	−	−	545	543	−
^5^D_0_→^7^F_0_ Eu^3+^	574	**573** *	574, 578	** 574 **	** 572 **	572, 578
^5^D_0_→^7^F_1_ Eu^3+^	586, 592, 603	600	588, 593, 597	593, 600	599	586, 593, 600
^5^D_0_→^7^F_2_ Eu^3+^	613, **625**	611, 626	**613**, 624	612, 624	611, 625	**612**, 618, 625
^5^D_0_→^7^F_3_ Eu^3+^	655	−	655	655	−	655
^5^D_0_→^7^F_4_ Eu^3+^	693, 704	694, 704	693, 705	694, 705	691, 703	693, 705
^5^D_0_→^7^F_5_ Eu^3+^	−	750	−	−	−	−

*—the most intense band in the spectra is underlined and in bold.

**Table 3 materials-16-05344-t003:** CIE chromaticity coordinates and correlated color temperature (CCT) for BSBF: Tb^3+^, BSBF: Eu^3+^, and BSBF: Eu^3+^, Tb^3+^, Ce^3+^ crystals.

λ_ex_, nm	CIE Coordinates		CCT, K	
	77 K	300 K	77 K	300 K
Ba_3_Sr_4_(BO_3_)_4_F_2_: Tb^3 +^				
300	(1) (0.345; 0.530)	(2) (0.344; 0.524)	5263	5278
Ba_3_Sr_4_(BO_3_)_4_F_2_: Eu^3 +^				
300	(3) (0.617; 0.378)	(4) (0.617; 0.378)	1836	1831
370	(5) (0.570; 0.406)	(6) (0.585; 0.410)	1756	1732
395	(7) (0.634; 0.359)	(8) (0.633; 0.359)	2185	2176
Ba_3_Sr_4_(BO_3_)_4_F_2_: Eu^3+^, Tb^3+^, Ce^3+^				
300	(9) (0.484; 0.349)	(10) (0.538; 0.373)	1822	1851
370	(11) (0.399; 0.439)	(12) (0.481; 0.452)	3946	2728
395	(13) (0.513; 0.327)	(14) (0.477;0.291)	6665	4849

## Data Availability

Data are currently unavailable (no publicly archived datasets created) but can be sent if anybody is interested in.

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
