# Peer review of "A Photoluminescence Study of Eu3+, Tb3+, Ce3+ Emission in Doped Crystals of Strontium-Barium Fluoride Borate Solid Solution Ba4−xSr3+x(BO3)4−yF2+3y (BSBF)"

_materials, 2023, doi:10.3390/ma16155344_

Round 1

Reviewer 1 Report

This work introduced a comprehensive study of photoluminescence of BSBF with different dopant like Eu3+, Tb3+ and Ce3+. Photoluminescence and photoluminescence excitation spectra are carefully researched. Origins of peaks are explained, temperature impact on photoluminescence are discussed.

This paper is well written and organized, may be accepted after minor revision as follow,

1.     This manuscript made comprehensive study about photoluminescence of BSBF. Other methods like Raman, TEM, XRD may also help us get a better understanding of samples. If it’s convenient for authors to add related data, it can provide other perspectives to study the material. If not, as this paper focuses on photoluminescence of BSBF, it’s also fine.

2.     For figure 2b, intensity of peak at 460nm decrease first and then bounce and finally fall again. Can authors explain this phenomenon?

3.     For figure 2a, will the intensity ratio of two peaks near 420 and 440 nm change when excitation energy change?

4.     Can authors provide more temperature dependent photoluminescence spectra as only two temperature 77K and 300K are hard to analysis temperature effect?

Reviewer 2 Report

- The authors have to reconsider the introduction and to extend and clarify the problem addressed and its importance, because their claim” The present study is aimed at unveiling the luminescence potential of Ba4–xSr3+x(BO3)4–49 yF2+3crystals doped with Eu3+, Tb3+, and Ce3+.” is too short and not convincing.

- The preferential substitution of alkali earth cations by Eu3+ should take place on Sr2+ site is well discussed and the luminescence analysis, too. However, there is no mention about the substitution of the Tb3+ and Ce3+…A proper discussion/comments is needed

- The conclusion has to include a section with the main conclusion about their study and the novelty and contribution in the field.

Minor editing of English language

Reviewer 3 Report

The current work studies Ce, Tb and Eu singly or co-doped BSBF crystals. Contents are interesting, and I can recommend a publication after some minor modifications listed below.

1.       In the crystal growth, detailed information is required. Do you use just melt and cooling method without any rotation, pulling-up/down and so on? How is the melting temperature? 2 °C per day for 15 days = RT + 30 °C? Judging from chemical composition, I cannot believe such a low melting temperature. How is the atmosphere?

2.       At line 80, what are squares in chemical reaction formula? Do you mean a vacancy?

3.       Would you like to comment the role of Ce? Is it a sensitizer?

4.       Throughout the entire work, information of emission intensity/efficiency is lacking. If possible, would you like to comment photoluminescence quantum yield or luminance? Otherwise, please compare with common commercial phosphors.

Reviewer 4 Report

Dear authors,

My primary concern revolves around the complete overlap of the luminescence spectra at the two distinct temperatures. I would appreciate if you could verify this result and also provide a comment and some references regarding similar materials with comparable behaviors.

I'm unsure about the scheme presented in line 80 concerning the vacancies.

Please make the necessary corrections in figures 3, 4, and 5 by adding Kelvin units to the legends.

Additionally, modify the colors in figure 3d.

Could you provide an explanation as to why the emission spectrum at 77K is absent in figure 4a?

Minor editing of English language required

Reviewer 5 Report

The present draft ‘A photoluminescence study of Eu3+, Tb3+, Ce3+ emission in 2 doped crystals of strontium-barium fluoride borate solid solution Ba4–xSr3+x(BO3)4–yF2+3y (BSBF)’ is well supported with the literature and would suggest accepting after the following corrections.

1.       The introduction is don’t highlight the significance of the work. I would suggest the authors rewrite and highlight the key research problem that they are trying to address.

2.       Line 93 ‘lend support to the view that preferential substitution of alkali earth cations by Eu3+ should take place in Sr2+ site’ -can the author explain how the charge is balanced or it is electronic doping?

3.       Line 148: ‘Such 0-0 transitions 148 are forbidden according to the  J selection rule’ Correct the typo. Same typo at line 170.

4.       Can the author measure the photoluminescence quantum yield for the system and compare the values?

5. The PL analysis is too busy with values and difficult to comprehend. All the analysis is supported by literature and does not bring any new information. I would advise you to shorten that and analyze them in table format. 

The language can be improved.

Round 2

Reviewer 2 Report

The authors provided proper answers to the comments.

Reviewer 5 Report

Author's has addressed all the questions and I would advise to accept in the present draft.